# A Simple Data Augmentation for Feature Distribution Skewed Federated Learning

## Abstract

Federated learning (FL) facilitates collaborative learning among multiple clients in a distributed manner and ensures privacy protection. However, its performance inevitably degrades, while suffering from data heterogeneity, *i.e.*, non-IID data. In this paper, we focus on the feature distribution skewed FL scenario, which is a common setting in real-world applications. The main challenge of this scenario is feature shift, which is caused by the different underlying distributions of local datasets. Although the previous attempts achieved impressive progress, few studies pay attention to the data itself, *i.e.* the root of this issue. To this end, the primary goal of this paper is to develop a general data augmentation technique at the input level, to mitigate the feature shift problem. To achieve this goal, we propose a simple yet remarkably effective data augmentation method, namely FedRDN, for feature distribution skewed FL, which randomly injects the statistics of the dataset from the entire federation into the client's data. Then, our method can effectively improve the generalization of features, and thereby mitigate the feature shift problem. Moreover, our FedRDN is a plug-and-play component, which can be seamlessly integrated into the data augmentation flow with only a few lines of code. Extensive experiments on several datasets show that the performance of various representative FL works can be further improved by integrating our FedRDN, which demonstrates its strong scalability and generalizability. The source code will be released.

## 1 Introduction

Federated learning (Li et al., 2020a) (FL) has been a de facto solution for distributed learning from discrete data among different edge devices like mobile phones, which has attracted wide attention from various communities (Peng et al.; Lee et al., 2021; Fang & Ye, 2022; Dong et al., 2022). It utilizes a credible server to communicate the privacy irrelevant information, *e.g.*, model parameters, thereby collaboratively training the model on multiple distributed data clients, while the local data of each client is invisible to others. Such a design is simple yet can achieve superior performance. However, it is inevitable to suffer data heterogeneity when we deploy FL in real-world applications, which will greatly degrade the performance of FL (Karimireddy et al., 2020; Li et al., 2020b; Luo et al., 2021).

Since data heterogeneity is widely common in many real-world cases, many researchers try to address this issue and improve the practicability of FL (Karimireddy et al., 2020; Li et al., 2020b; Hsu et al., 2019; Reisizadeh et al., 2020). However, most of them try to design a robust FL method that can handle different data heterogeneity settings. In fact, such a solution is suboptimal, due to the discrepancy between different data heterogeneity settings, such as feature distribution skew and label distribution skew (Li et al., 2022). Hence, some studies try to design special FL algorithms for different settings (Zhang et al., 2022; Zhou & Konukoglu, 2023). In this work, we focus on solving the feature distribution skew problem, a typical data statistical heterogeneity scenario in FL, as the data from discrete clients are always collected from different devices or environments, incurring different underlying distributions among clients' data. In FL, the client-side only trains the data on its skewed data, resulting in significant local model bias, and a typical manifestation is inconsistent feature distribution across different clients, namely ***feature shift*** (Li et al., 2021d; Zhou & Konukoglu, 2023). To address this problem, FedBN (Li et al., 2021d) learns the client-specific batch normalization layer parameters for each client instead of learning only a single global model.

HarmoFL (Jiang et al., 2022) utilizes the potential knowledge of medical images in the frequency domain to mitigate the feature shift problem.

While the feature distribution skew has been explored in the area of FL, few attention has been paid to the data itself. In FL, the common practice (Jiang et al., 2022; Zhou & Konukoglu, 2023; Li et al., 2021d) is to integrate some traditional data operations in the data augmentation flow (*e.g.*, *transforms.Compose*() in Pytorch) of each client like horizontal and vertical flips. Even though this setting has shown effectiveness and generalizability in centralized learning, it overlooks the effectiveness of data augmentation for FL at the input level, *i.e.*, data augmentation flow, which leaves an alternative space to explore. Since the root cause of feature shift problem lies in the divergence of data distributions, we ask two questions: 1) ***Why not proactively resolve it by directly processing the data?*** 2) ***Can we design an FL-specific data augmentation operation that can be integrated into the data augmentation flow?*** To answer these questions, we try to design a plug-and-play input-level data augmentation technique.

Although it may appear straightforward, effectively implementing data augmentation in federated learning poses a significant challenge, due to the lack of direct access to the external data of other users. Hence, how to inject global information into augmented samples and thereby mitigate the distribution bias between different local datasets is the core of the challenge. In this regard, Fed-Mix (Yoon et al., 2021) extends Mixup to federated learning for label distribution skew by incorporating the mixing of averaged data across multiple clients. However, it is only suitable for the classification task and has not yet demonstrated the effectiveness for feature distribution skew. Furthermore, permitting the exchange of averaged data introduces certain privacy concerns and risks. In more recent study, a data augmentation approach, namely FedFA (Zhou & Konukoglu, 2023), is proposed for feature distribution skew, which mitigates the feature shift problem through feature augmentation based on the statistics of latent features. Even though, the approach applies augmentation at the feature level, which lacks versatility and cannot be seamlessly integrated into a comprehensive data augmentation pipeline.

In this work, we propose a novel federated data augmentation technique, called FedRDN. We argue that the local bias is rooted in the model trained on the limited and skewed distribution. In centralized learning, we can collect data of many different distributions to learn generalized feature representations. This motivates us to augment the data to multiple local distributions, which indirectly reduces the difference between data from different distributions. To achieve that in the setting of FL, our FedRDN extends the standard data normalization to FL by randomly injecting statistics of local datasets into augmented samples, which is based on the insight that statistics of data are the essential characteristic of data distribution. It enables clients to access a wider range distribution of data, and thereby enhances the generalizability of features.

Our FedRDN is a remarkably simple yet surprisingly effective method. It is a non-parametric approach that incurs minimal additional computation and communication overhead, seamlessly integrating into the data augmentation pipeline with only a few lines of code. After employing our method, significant improvements have been observed across various typical FL methods. In a nutshell, our contributions are summarized as follows:

- We explore the input-level data augmentation technique for feature distribution skewed FL, which gives more insights into how to understand and solve this problem.

- We propose a novel plug-and-play data augmentation technique, FedRDN, which can be easily integrated into the data augmentation pipeline to mitigate the feature shift for feature distribution skewed FL.

- We conduct extensive experiments on two classification datasets and an MRI segmentation dataset to demonstrate the effectiveness and generalization of our method, *i.e.*, it outperforms traditional data augmentation techniques and improves the performance of various typical FL methods.

## 2 RELATED WORK

**Federated Learning with Statistical Heterogeneity**     FL allows multiple discrete clients to collaborate in training a global model while preserving privacy. The pioneering work, FedAvg (McMahan et al., 2017), is the most widely used FL algorithm, yielding success in various applications.

However, the performance of FedAvg is inevitably degraded when suffering data statistical heterogeneity (Li et al., 2020b; Karimireddy et al., 2020), *i.e.*, non-IID data, posing a fundamental challenge in this field. To address this, a variety of novel FL frameworks have been proposed, with representative examples such as FedProx (Li et al., 2020b), Scaffold (Karimireddy et al., 2020), and FedNova (Wang et al., 2020). They improved FedAvg by modifying the local training (Li et al., 2021c; Acar et al.) or model aggregation (Yurochkin et al., 2019; Lin et al., 2020) to increase stability in heterogeneous environments. Despite the progress, most of them ignore the differences in various data heterogeneity, like the heterogeneity of label distribution skew and feature distribution skew lies in label and images (Li et al., 2020a), respectively. Therefore, recent studies attempt to more targeted methods for different data heterogeneity. For example, FedLC (Zhang et al., 2022) addressed the label distribution skew by logits calibration. As for feature distribution skew, the target of this work, FedBN (Li et al., 2021d) learned the heterogeneous distribution of each client by personalized BN parameters. In addition, HarmoFL (Jiang et al., 2022) investigated specialized knowledge of frequency to mitigate feature shifts in different domains. Although achieving significant progress, they only focus on mitigating the feature shift at the model optimization or aggregation stage, neglecting the data itself which is the root of the feature shift. Different from them, we aim to explore a new data augmentation technique to address this issue. Since operating on the data level, our method can be combined with various FL methods to further improve their performance, which shows stronger generalizability.

**Data Augmentation** Data augmentation (Kukačka et al., 2017; Kumar et al., 2023) is a widely used technique in machine learning, which can alleviate overfitting and improve the generalization of the model. For computer vision tasks, the neural networks are typically trained with a data augmentation flow containing various techniques like random flipping, cropping, and data normalization. Different from these early label-preserving techniques (Zhong et al., 2020), label-perturbing techniques are recently popular such as MIXUP (Zhang et al.) and CUTMIX (Yun et al., 2019). It augments samples by fusing not only two different images but also their labels. Except for the above input-level augmentation techniques, some feature-level augmentation techniques (Li et al., 2021a;b; Venkataramanan et al., 2022) that make augmentation in feature space have achieved success. Recently, some studies start to introduce data augmentation techniques into FL. For example, FedMix (Yoon et al., 2021) proposed a variant of MIXUP in FL, which shows its effectiveness in label distribution skewed setting. However, it has not been demonstrated the effectiveness to address feature distribution skew. More importantly, FedMix requires sharing averaged local data which increases the risk of privacy. For feature distribution skew, FedFA (Zhou & Konukoglu, 2023) augmented the features of clients by the statistics of features. However, it operates on the feature level, thereby cannot be treated as a plug-and-play component. Different from FedFA, our proposed augmentation method operates on the input level, which can be seamlessly integrated into the data augmentation flow, showing stronger scalability. In addition, our method can further improve the performance of FedFA due to operating in different spaces.

## 3 METHODOLOGY

### 3.1 PRELIMINARIES

**Federated Learning** Supposed that a federated learning system is composed of $K$ clients $\{C_1, C_2, \ldots, C_K\}$ and a central server. For client $C_k$ ($k \in [K]$), there are $n_k$ supervised training samples $\{x_i, y_i\}_{i=1}^{n_k}$, where image $x_i$ and label $y_i$ from a joint distribution $(x_i, y_i) \sim P_k(x, y)$. Besides, each client trains a local model $f(\boldsymbol{w}_k)$ only on its private dataset. The goal of federated learning is to learn a global model by minimizing the summation empirical risk of each client:

$$\min \mathcal{L} = \sum_{k=1}^{K} \gamma_k \mathcal{L}_k, \quad \text{where} \quad \gamma_k = \frac{n_k}{\sum_{i=1}^{K} n_i}. \tag{1}$$

To achieve this goal, the leading method FedAvg (McMahan et al., 2017) performs $E$ epochs local training and then averages the parameters of all local models to get the global model at each communication round $t \in [T]$, which can be described as:

$$\boldsymbol{w}_G^{t+1} = \sum_{k=1}^{K} \gamma_k \boldsymbol{w}_k^t. \tag{2}$$

**Feature Distribution Skew**    The underlying data distribution $P_k(x, y)$ can be rewritten as $P_k(y|x)P_k(x)$, and $P_k(x)$ varies across clients while $P_k(y|x)$ is consistent for all clients. Moreover, the different underlying data distributions will lead to the inconsistent feature distribution of each client, thereby degrading the performance of the global model.

**Data Normalization**    Normalization is a popular data preprocess operation, which can transform the data distribution to standard normal distribution. In detail, given an $C$-channel image $x \in \mathbb{R}^{C \times H \times W}$ with spatial size $(H \times W)$, it transforms image as:

$$\hat{x} = \frac{x - \mu}{\sigma}, \quad \mu, \sigma \in \mathbb{R}^C, \tag{3}$$

where $\mu$ and $\sigma$ are channel-wise means and standard deviation, respectively, and they are usually manually set in experiential or statistical values from the real dataset.

### 3.2 FedRDN: Federated Random Data Normalization

In this section, we present the detail of the proposed Federated Random Data Normalization (FedRDN) method. Different from the previous FL works, FedRDN focuses on mitigating the distribution discrepancy at the data augmentation stage, thereby it can be easily plugged into various FL works to improve their performance. The goal of the FedRDN is to let each client learn as many distributions as possible instead of self-biased distribution, which is beneficial to feature generalization. To achieve this, it performs implicit data augmentation by manipulating multiple-clients channel-wise data statistics during training at each client. We will introduce the detail of our method in the following.

**Data Distribution Statistic** The approximate distribution of the data can be estimated using statistical methods. Therefore, we can obtain an approximate distribution by computing the statistics of the local dataset, *i.e.*, $P_k \sim \mathcal{N}(\mu^k, (\sigma^k)^2)$, where $\mu^k$ and $\sigma^k$ are mean and standard deviation, respectively. Specifically, to estimate such underlying distribution information of each client, we compute the channel-wise statistics within each local dataset in client-side before the start of training:

$$\mu^k = \sum_{i=1}^{n_k} \mu_i^k \in \mathbb{R}^C, \quad \sigma^k = \sum_{i=1}^{n_k} \sigma_i^k \in \mathbb{R}^C, \tag{4}$$

where $\mu_i^k$ and $\sigma_i^k$ are sample-level channel-wise statistics, and they can be computed as:

$$\mu_i^k = \frac{1}{HW} \sum_{h=1}^{H} \sum_{w=1}^{W} x_i^{k,(h,w)}, \quad \sigma_i^k = \sqrt{\frac{1}{HW} \sum_{h=1}^{H} \sum_{w=1}^{W} (x_i^{k,(h,w)} - \mu_i^k)^2}, \tag{5}$$

where $x_i^{k,(h,w)}$ represents the image pixel at spatial location $(h, w)$. Following this, all data distribution statistics will be sent to the server and aggregated by the server. The aggregated statistics $\{(\mu^k, \sigma^k)\}_{k=1}^K$ are shared among clients.

**Data Augmentation at Training Phase**    After obtaining the statistical information of each client, we utilize them to augment data during training. Considering an image $x_i^k$, different from the normal data normalization that transforms the image according to a fixed statistic, we transform the image by randomly selecting the mean and standard deviation from statistics $\{(\mu^k, \sigma^k)\}_{k=1}^K$, which can be described as:

$$\hat{x}_i^k = \frac{x_i^k - \mu^j}{\sigma^j}, \quad (\mu^j, \sigma^j) \sim \{(\mu^k, \sigma^k)\}_{k=1}^K. \tag{6}$$

Notably, the statistic $(\mu^j, \sigma^j)$ will be randomly reselected for each image at each training epoch. Therefore, the images will be transformed into multiple distributions after several epochs of training. In this way, we seamlessly inject global information into augmented samples. The local model can learn the distribution information of all clients, thereby making the learned features more generalized.

**Data Augmentation at Testing Phase**    Random selection will lead to the uncertainty of the prediction. If we randomly select statistics during testing, just as we did during the training phase, the output results may differ due to the varied statistics chosen. Since the clients have learned from

multiple distributions, we only need to select the corresponding statistics of each client to ensure consistent and accurate evaluation during testing. The above operation can be written as:

$$\hat{x}_i^k = \frac{x_i^k - \mu^k}{\sigma^k}. \tag{7}$$

The overview of two processes, *i.e.*, statistic computing and data augmentaion, are presented in Algorithm 1 and 2.

---

**Algorithm 1:** Compute Statistic

**Input:** $K$ datasets: $\{P_1, P_2, \ldots, P_K\}$

1 **for** *client $k = 1, 2, ..., K$ **parallelly** do*
2     **for** $x_i^k \sim P_k$ **do**
3        $\mu_i^k = \frac{1}{HW} \sum_{h=1}^{H} \sum_{w=1}^{W} x_i^{k,(h,w)}$
4        $\sigma_i^k =$
         $\sqrt{\frac{1}{HW} \sum_{h=1}^{H} \sum_{w=1}^{W} (x_i^{k,(h,w)} - \mu_i^k)^2}$
5     **end**
6     $\mu^k = \sum_{i=1}^{n_k} \mu_i^k, \quad \sigma^k = \sum_{i=1}^{n_k} \sigma_i^k$
7 **end**
8 **Return** $\{(\mu^k, \sigma^k)\}_{k=1}^K$

---

**Algorithm 2:** Data Augmentation

**Input:** $K$ datasets: $\{P_1, P_2, \ldots, P_K\}$,
       Data Statistics $\{(\mu^k, \sigma^k)\}_{k=1}^K$

1 **for** $x_i^k \sim P_k$ **do**
2     **if** *is Train* **then**
3        $(\mu^j, \sigma^j) \sim \{(\mu^k, \sigma^k)\}_{k=1}^K$
4     **else**
5        $(\mu^j, \sigma^j) = (\mu^k, \sigma^k)$
6     **end**
7     $\hat{x}_i^k = \frac{x_i^k - \mu^j}{\sigma^j}$
       `// training or testing`
8 **end**

---

**Privacy Security** The previous input-level augmentation method, *i.e.*, FedMix (Yoon et al., 2021), shares the average images per batch, leading to the increased risk of privacy. Different from it, our method only shares the privacy irrelevant information, *i.e.*, dataset-level mean and standard deviation. In addition, we can not reverse the individual image from the shared information because it is statistical information of the whole dataset. Therefore, our method has a high level of privacy security.

## 4 EXPERIMENTS

### 4.1 EXPERIMENTAL SETUP

**Datasets** We conduct extensive experiments on three real-world datasets: **Office-Caltech-10** (Gong et al., 2012), **DomainNet** (Peng et al., 2019), and **ProstateMRI** (Liu et al., 2020), which are widely used in feature distribution skewed FL settings (Li et al., 2021d; Zhou & Konukoglu, 2023; Jiang et al., 2022). There are two different tasks including image classification (Office-Caltech-10, DomainNet) and medical image segmentation (ProstateMRI). Following previous work (Li et al., 2021d; Zhou & Konukoglu, 2023), we employ the subsets as clients when conducting experiments on each dataset.

**Baselines** To demonstrate the effectiveness of our method, we build four different data augmentation flows: one flow has some basic data augmentation techniques like random flipping, one flow adds the conventional normalization technique, another flow adds the FedMix (Yoon et al., 2021) data augmentation technique, and the rest integrates our proposed augmentation method into basic data augmentation flow. Since it is infeasible to deploy FedMix into segmentation tasks, we only utilize it for image classification tasks. Following, we integrate them into different typical FL methods. In detail, we employ seven state-of-the-art FL methods to demonstrate the generalizability of our method, including **FedAvg** (McMahan et al., 2017), **FedAvgM** (Hsu et al., 2019), **FedProx** (Li et al., 2020b), **Scaffold** (Karimireddy et al., 2020), **FedNova** (Wang et al., 2020), **FedProto** (Tan et al., 2022), and **FedFA** (Zhou & Konukoglu, 2023) for validation of image classification task. Moreover, we select four of them which are general for different tasks to validate the effectiveness of our method on medical image segmentation tasks. To quantitatively evaluate the performance, we utilize the top-1 accuracy for image classification while the medical segmentation is evaluated with Dice coefficient. Notably, there are some important hyper-parameters for some FL methods. For instance, the FedProx and FedProto have $\mu$ to control the contribution of an additional loss function. We empirically set $\mu$ to 0.001 for FedProx and 1 for FedProto on all datasets. Besides, FedAvgM has a momentum hyper-parameter to control the momentum update of the global model parameters,

Table 1: **The test accuracy (%) of all approaches on office-Caltech-10 (**Gong et al., 2012**)
and DomainNet (**Peng et al., 2019**).** For a detailed comparison, we present the test accuracy of
each client *i.e.*, **Office-Caltech-10**: A(Amazon), C(Caltech), D(DSLR), W(Webcam), **DomainNet**:
C(Clipart), I(Infograph), P(Painting), Q(Quickdraw), R(Real), S(Sketch), and the average result. ↑
and ↓ show the rise and fall of the average result before and after augmentation. We mark our results
in bold. (norm.: conventional data normalization)

| Method | Office-Caltech-10 (Gong et al., 2012) | | | | | DomainNet (Peng et al., 2019) | | | | | | |
|---|---|---|---|---|---|---|---|---|---|---|---|---|
| | A | C | D | W | .Avg | C | I | P | Q | R | S | .Avg |
| FedAvg | 53.12 | 44.88 | 65.62 | 86.44 | 62.51 | 50.38 | 22.83 | 36.99 | 58.10 | 46.09 | 39.53 | 42.32 |
| FedAvg + *norm* | 50.52 | 43.55 | 68.75 | 83.05 | $61.46_{(1.05)}$ ↓ | 48.28 | 23.28 | 37.80 | 54.20 | 48.97 | 41.69 | $42.37_{(0.05)}$ ↑ |
| FedAvg + *FedMix* | 49.47 | 41.77 | 75.00 | 88.13 | $63.59_{(1.08)}$ ↑ | 48.66 | 23.43 | 38.12 | 55.10 | 49.46 | 41.33 | $42.68_{(0.36)}$ ↑ |
| FedAvg + *FedRDN* | **60.93** | **45.77** | **84.37** | **88.13** | **$69.80_{(7.29)}$** ↑ | **48.85** | 22.67 | **39.41** | **60.30** | 49.46 | 40.61 | **$43.55_{(1.23)}$** ↑ |
| FedProx | 53.12 | 45.33 | 62.50 | 86.44 | 61.84 | 52.66 | 23.89 | 35.21 | 56.70 | 46.75 | 41.87 | 42.85 |
| FedProx + *norm* | 51.04 | 45.77 | 68.75 | 84.74 | $62.57_{(0.73)}$ ↑ | 47.14 | 24.35 | 34.57 | 59.60 | 44.86 | 38.98 | $41.58_{(1.27)}$ ↓ |
| FedProx + *FedMix* | 47.39 | 38.66 | 78.12 | 91.52 | $63.92_{(2.08)}$ ↑ | 47.90 | 22.37 | 37.31 | 53.90 | 48.47 | 43.14 | $42.18_{(0.67)}$ ↓ |
| FedProx + *FedRDN* | **61.45** | 44.88 | **84.37** | **88.13** | **$69.71_{(7.87)}$** ↑ | 50.57 | **24.96** | 38.77 | **61.20** | **51.35** | 40.97 | **$44.63_{(1.78)}$** ↑ |
| FedNova | 50.00 | 42.22 | 62.50 | 88.13 | 60.71 | 51.71 | 23.74 | 38.77 | 56.20 | 45.52 | 38.44 | 42.39 |
| FedNova + *norm* | 52.08 | 45.33 | 68.75 | 86.44 | $63.15_{(2.44)}$ ↑ | 49.23 | 24.35 | 34.24 | 55.80 | 45.52 | 42.23 | $41.90_{(0.49)}$ ↓ |
| FedNova + *FedMix* | 48.95 | 42.66 | 78.12 | 83.05 | $63.20_{(2.49)}$ ↑ | 47.90 | 24.04 | 36.67 | 59.10 | 46.67 | 42.41 | $42.80_{(0.41)}$ ↑ |
| FedNova + *FedRDN* | **63.02** | 41.33 | **84.37** | **89.83** | **$69.63_{(8.71)}$** ↑ | 50.57 | 23.43 | **40.22** | **57.30** | **51.84** | **40.43** | **$43.96_{(1.57)}$** ↑ |
| Scaffold | 52.60 | 42.66 | 53.12 | 81.35 | 57.43 | 46.95 | 22.83 | 34.57 | 46.50 | 47.00 | 40.97 | 39.80 |
| Scaffold + *norm* | 46.87 | 40.00 | 59.37 | 86.44 | $58.17_{(0.74)}$ ↑ | 47.33 | 22.83 | 33.11 | 58.30 | 46.01 | 42.05 | $41.61_{(1.81)}$ ↑ |
| Scaffold + *FedMix* | 52.08 | 40.88 | 75.00 | 89.83 | $64.45_{(7.02)}$ ↑ | 45.24 | 23.28 | 34.73 | 47.50 | 44.78 | 40.97 | $39.42_{(0.38)}$ ↓ |
| Scaffold + *FedRDN* | **65.10** | 41.77 | **81.25** | 86.44 | **$68.64_{(11.21)}$** ↑ | **51.52** | **23.89** | **37.96** | **56.20** | **48.97** | 38.80 | **$42.89_{(3.09)}$** ↑ |
| FedAvgM | 48.43 | 45.33 | 62.50 | 83.05 | 59.83 | 45.81 | 22.52 | 37.96 | 50.10 | 48.23 | 41.87 | 41.08 |
| FedAvgM + *norm* | 51.04 | 44.88 | 62.50 | 86.44 | $61.21_{(1.38)}$ ↑ | 46.95 | 24.96 | 35.86 | 49.70 | 45.43 | 40.43 | $40.55_{(0.53)}$ ↓ |
| FedAvgM + *FedMix* | 50.00 | 41.77 | 65.62 | 83.05 | $60.11_{(0.29)}$ ↑ | 48.28 | 25.87 | 40.06 | 51.50 | 48.56 | 38.62 | $42.15_{(1.07)}$ ↑ |
| FedAvgM + *FedRDN* | **62.50** | 43.11 | **84.37** | **88.13** | **$69.53_{(9.70)}$** ↑ | 48.09 | 22.98 | **41.03** | **63.80** | **49.79** | 38.08 | **$43.96_{(2.88)}$** ↑ |
| FedProto | 55.72 | 44.44 | 68.75 | 86.44 | 63.84 | 48.28 | 25.11 | 35.86 | 51.30 | 43.79 | 37.18 | 40.25 |
| FedProto + *norm* | 53.64 | 44.88 | 56.25 | 86.44 | $60.30_{(3.54)}$ ↓ | 45.81 | 23.43 | 35.70 | 58.30 | 45.27 | 40.79 | $41.55_{(1.30)}$ ↑ |
| FedProto + *FedMix* | 53.64 | 41.77 | 84.37 | 88.13 | $66.98_{(3.14)}$ ↑ | 47.33 | 23.43 | 37.47 | 52.70 | 44.94 | 42.41 | $41.38_{(1.13)}$ ↑ |
| FedProto + *FedRDN* | **66.14** | **46.22** | 84.37 | **89.83** | **$71.64_{(7.80)}$** ↑ | **49.42** | 22.37 | **41.51** | **57.90** | **51.43** | 38.44 | **$43.51_{(3.26)}$** ↑ |
| FedFA | 60.93 | 48.44 | 81.25 | 89.83 | 70.11 | 48.09 | 23.74 | 39.58 | 64.20 | 48.06 | 43.14 | 44.47 |
| FedFA + *norm* | 60.93 | 50.66 | 81.25 | 84.74 | $69.40_{(0.71)}$ ↓ | 49.04 | 24.20 | 38.28 | 60.70 | 45.93 | 42.59 | $43.46_{(1.01)}$ ↓ |
| FedFA + *FedMix* | 56.25 | 48.44 | 84.37 | 88.13 | $69.30_{(0.81)}$ ↓ | 45.81 | 22.52 | 33.11 | 51.10 | 43.13 | 37.90 | $38.93_{(5.54)}$ ↓ |
| FedFA + *FedRDN* | **62.50** | **48.88** | **90.62** | **91.52** | **$73.38_{(3.27)}$** ↑ | **52.85** | **24.50** | 37.80 | 61.90 | **50.45** | 42.59 | **$45.01_{(0.54)}$** ↑ |

which is set to 0.9 for two image classification datasets and 0.01 for ProstateMRI. FedMix also has
two hyper-parameters, *i.e.*, the batch of mean images and λ to control images fusing. We adopted
the best configuration from the original paper.

**Network Architecture** Following previous work (Li et al., 2021d; Zhou & Konukoglu, 2023),
we employ the AlexNet (Krizhevsky et al., 2017) as the image classification model and the U-
Net (Ronneberger et al., 2015) as the medical image segmentation model.

**Implementation Details** All methods are implemented by PyTorch, and we conduct all experi-
ments on a single NVIDIA GTX 1080Ti GPU with 11GB of memory. The batch size is 32 for two
image classification datasets and 16 for ProstateMRI dataset. We adopt the SGD optimizer with
learning rate 0.01 and weight decay 1e-5 for image classification datasets, and the Adam optimizer
with learning rate 1e-3 and weight decay 1e-4 for ProstateMRI. Furthermore, we run 100 communi-
cation rounds on image classification tasks while the number of rounds for the medical segmentation
task is 200, and each round has 5 epochs of local training. More importantly, for a fair comparison,
we train all methods in the same environment and ensure that all methods have converged. Due to
the page limitation, we provide additional experimental results in Appendix A.

Table 2: **The dice score (%) of all approaches on ProstateMRI (Liu et al., 2020)**. For a detailed comparison, we present the test result of six clients: BIDMC, HK, I2CVB, BMC, RUNMC, UCL, and the average result. ↑ and ↓ show the rise and fall of average result before and after augmentation. We emphasize our result in bold. (norm.: conventional data normalization)

| Method | ProstateMRI (Liu et al., 2020) | | | | | | |
| --- | --- | --- | --- | --- | --- | --- | --- |
| | BIDMC | HK | I2CVB | BMC | RUNMC | UCL | .Avg |
| FedAvg | 84.16 | 94.51 | 94.60 | 88.43 | 92.78 | 52.65 | 90.02 |
| FedAvg + *norm* | 86.20 | 92.53 | 94.74 | 89.85 | 92.18 | 87.91 | 90.57$_{(0.55)}$ ↑ |
| FedAvg + *FedRDN* | **89.34** | **94.41** | **93.85** | **91.46** | **94.19** | **90.65** | **92.32**$_{(2.30)}$ ↑ |
| FedProx | 84.47 | 94.60 | 94.87 | 90.29 | 92.72 | 86.60 | 90.59 |
| FedProx + *norm* | 84.47 | 94.48 | 95.06 | 88.79 | 92.90 | 85.35 | 90.18$_{(0.41)}$ ↓ |
| FedProx + *FedRDN* | **88.87** | **94.19** | **95.09** | **90.99** | **93.03** | **89.17** | **91.89**$_{(1.30)}$ ↑ |
| FedAvgM | 87.02 | 94.32 | 94.29 | 91.35 | 92.83 | 86.75 | 91.09 |
| FedAvgM + *norm* | 89.05 | 93.59 | 94.75 | 89.93 | 93.52 | 88.22 | 91.51$_{(0.42)}$ ↑ |
| FedAvgM + *FedRDN* | 88.37 | **94.67** | **95.40** | **90.40** | **93.28** | **88.39** | **91.75**$_{(0.66)}$ ↑ |
| FedFA | 89.18 | 92.77 | 94.18 | 92.62 | 93.63 | 89.04 | 91.90 |
| FedFA + *norm* | 89.12 | 94.40 | 95.22 | 91.95 | 93.42 | 89.28 | 92.23$_{(0.33)}$ ↑ |
| FedFA + *FedRDN* | **91.81** | **94.65** | **95.67** | **92.37** | **94.33** | **90.19** | **93.14**$_{(1.24)}$ ↑ |

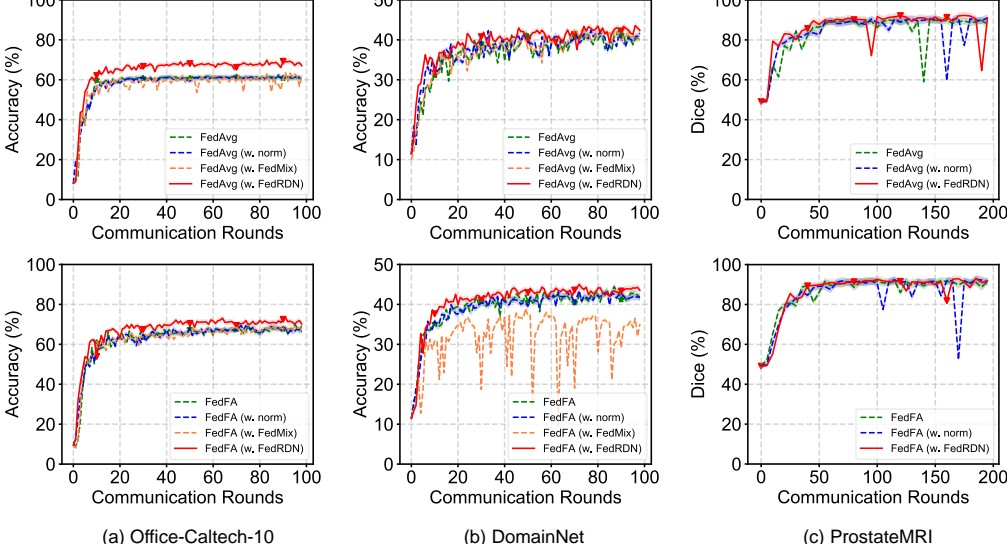

Figure 1: **Illustration of test performance versus communication rounds** on (a) Office-Caltech-10 (Gong et al., 2012), (b) DomainNet (Peng et al., 2019), and (c) ProstateMRI (Liu et al., 2020).

## 4.2 MAIN RESULTS

In this section, we present the overall results on three benchmarks: Office-Caltech-10 and Domain-Net in Table 1 and ProstateMRI in Table 2, including two different tasks, *i.e.*, image classification, and medical image segmentation. For a detailed comparison, we present the test accuracy of each client and the average result.

*All FL methods yield significant improvements combined with FedRDN consistently over three datasets.* As we can see, FedRDN leads to consistent performance improvement for all FL baselines across three benchmarks compared with using the base data augmentation flow. The improvements of FedRDN can be large as **11.21%** on Office-Caltech-10, **3.26%** on DomainNet, and **1.37%** on ProstateMRI, respectively. Especially for FedFA, the state-of-the-art FL method for feature distribution skewed FL setting, can still gain improvements, *e.g.*, **3.27%** on Office-Caltech-10. This indicates that input-level augmentation and feature-level augmentation are not contradictory and can

Table 3: **Generalization performance of local model** on Office-Caltech-10 (Gong et al., 2012). We mark the best result in bold.

| Source-site | Target-site | FedAvg | FedAvg + *norm* | FedAvg + *FedMix* | FedAvg + ***FedRDN(Ours)*** |
|---|---|---|---|---|---|
| Amazon | Caltech | 48.00 | 48.09 | 42.22 | **48.88** |
| | DSLR | 46.87 | 44.41 | 59.37 | **81.25** |
| | Webcam | 77.96 | 77.96 | **84.74** | 83.05 |
| Caltech | Amazon | 58.33 | 58.85 | 51.56 | **63.02** |
| | DSLR | 68.75 | 68.75 | 75.00 | **84.37** |
| | Webcam | 83.05 | 83.05 | **91.52** | 86.35 |
| DSLR | Amazon | 42.70 | 41.14 | 33.85 | **60.93** |
| | Caltech | 33.33 | 35.11 | 35.55 | **35.56** |
| | Webcam | 79.66 | 77.96 | 74.57 | **84.74** |
| Webcam | Amazon | 53.64 | 55.20 | 47.91 | **65.10** |
| | Caltech | 41.33 | 43.55 | 40.00 | **48.00** |
| | DSLR | 68.75 | 68.75 | 78.12 | **87.50** |

be used simultaneously. Moreover, some weaker FL methods can even achieve better performance than others when using FedRDN. For example, after using FedRDN, the accuracy of FedAvg can be significantly higher than all other FL methods except FedFA, and FedProto can be even higher than FedFA. The above results demonstrate the effectiveness of the data-level solution, which can effectively mitigate the feature shift. Besides, compared with other robust FL methods, our method has a stronger scalability and generalization ability.

***FedRDN is superior to other input-level data augmentation techniques.*** As shown in Table 1 and 2, FedRDN shows leading performance compared with conventional data normalization and FedMix, a previous input-level data augmentation technique. Besides, these two augmentation techniques can even decrease the performance of the method in several cases, while FedRDN can achieve consistent improvements. For instance, FedAvg and FedProto yield a drop as large as **1.05%** and **3.54%** with conventional data normalization on Office-Caltech-10, respectively. FedProx and FedFA show a drop as **0.67%** and **5.54%** on DomainNet, respectively, when they combined with FedMix. The above results demonstrated the effectiveness of FedRDN, and it has stronger generalizability compared with other data augmentation methods.

### 4.3 COMMUNICATION EFFICIENCY

**Convergence**    To explore the impact of various data augmentation techniques on the convergence, we draw the test performance curve of FedAvg and a state-of-the-art FL method, *i.e.*, FedFA, with different communication rounds on three datasets as shown in Fig. 1. Apparently, FedRDN will not introduce any negative impact on the convergence of the method and even yield a faster convergence at the early training stage ($0 \sim 10$ rounds) in some cases. As the training goes on, FedRDN achieves a more optimal solution. Besides, compared with other methods, the convergence curves of FedRDN are more stable.

**Communication Cost**    In addition to the existing communication overhead of FL methods, the additional communication cost in FedRDN is only for statistical information. The dimension of the statistic is so small (mean and standard deviation are $\mathbb{R}^3$ for RGB images), that the increased communication cost can be neglected. This is much different from the FedMix, which needs to share the average images per batch. The increased communication cost of FedMix is as large as 156MB on Office-Caltech-10 and 567MB on DomainNet while the batch size of averaged images is 5, even larger than the size of model parameters.

### 4.4 CROSS-SITE GENERALIZATION PERFORMANCE

As stated before, FedRDN augments the data with luxuriant distribution from all clients to learn the generalized model. Therefore, we further explore the generalization performance of local models by cross-site evaluation, and the results are presented in Table 3. As we can see, All local models of

Table 4: **The performance of FedRDN over FedRDN-V** on three datasets.

| Method | Office-Caltech-10 | DomainNet | **ProstateMRI** |
|---|---|---|---|
| FedAvg | 62.51 | 42.32 | 90.02 |
| FedAvg + *FedRDN-V* | 61.46 | 42.99 | 91.14 |
| FedAvg + *FedRDN* (ours) | **69.80** | **43.55** | **92.32** |

FedRDN yield a solid improvement compared with the FedAvg and our method shows better generalization performance compared to other methods. The above results demonstrate that FedRDN can effectively mitigate the domain shift between different local datasets, which is beneficial to model aggregation. This is the reason why our method works.

### 4.5 DIAGNOSTIC EXPERIMENT

**FedRDN vs. FedRDN-V**    To deeply explore FedRDN, we develop a variant, FedRDN-V. Instead of randomly transforming, it transforms the images with the average mean $\hat{u}$ and standard deviation $\hat{\sigma}$ of all clients during training and testing phases:

$$\hat{u} = \sum_{k=1}^{K} \mu_k, \quad \hat{\sigma} = \sum_{k=1}^{K} \sigma_k. \tag{8}$$

The results of comparison over three datasets are presented in Table 4. Apparently, FedRDN-V yields a significant drop compared with our method. This indicates the effectiveness of our method is not from the traditional data normalization but augmenting samples with the information from the multiple real local distributions. By this, each local model will be more generalized instead of biasing in skewed underlying distribution.

**Robust to Local Epochs**    To explore the robustness of FedRDN for different local epochs, we tune the local epochs from the {1, 5, 10, 15, 20} and evaluate the performance of the learned model. The results are presented in Fig. 2. Generally, more epochs of local training will increase the discrepancy under data heterogeneity, leading to slower convergence, which degrades the performance at the same communication rounds. The result of FedAvg validates this. By contrast, our method can obtain consistent improvements with different settings of local epochs. Moreover, our approach has stable performance across different local epoch settings due to effectively addressing the data heterogeneity.

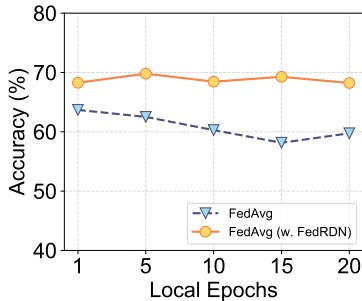

Figure 2: **Illustration of test performance versus local epochs** on Office-Caltech-10 (Gong et al., 2012).

## 5 CONCLUSION

In this paper, we focus on addressing the feature distribution skewed FL scenario. Different from the previous insights for this problem, we try to solve this challenge from the input-data level. The proposed novel data augmentation technique, FedRDN, is a plug-and-play component that can be easily integrated into the data augmentation flow, thereby effectively mitigating the feature shift. Our extensive experiments show that FedRDN can further improve the performance of various state-of-the-art FL methods across three datasets, which demonstrates the scalability, generalizability, and effectiveness of our method.

**Limitations**    This research provide a new direction to address the feature skew, *i.e.*, a data perspective. However, this work primarily focuses on visual tasks, where statistical quantities are privacy-agnostic information, as they only capture the distribution information instead of individual-level information. Considering the scalability, generalizability, and effectiveness of FedRDN, we believe this work contribute substantively to the ongoing discourse in the field of federated learning.

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

# A ADDITIONAL EXPERIMENTS

## A.1 FEATURE DISTRIBUTION

To yield more insights about FedRDN, we utilize T-SNE (Van der Maaten & Hinton, 2008) , a popular tool, to visualize the features of the global model before and after augmentation. Specifically, we visualize the features of test samples for each client and the results are shown in Fig. 3. Apparently, FedRDN can help learn a more consistent feature distribution for different clients, while the learned feature of FedAvg is obviously biased, *e.g.*, client 2 (DSLR). This validates what we previously stated: ***FedRDN is beneficial to learn more generalized features***.

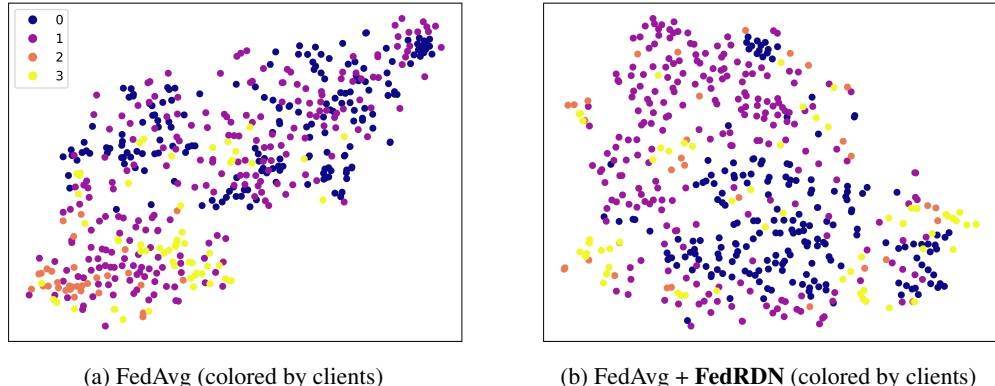

(a) FedAvg (colored by clients)      (b) FedAvg + **FedRDN** (colored by clients)

Figure 3: **T-SNE visualization of features** on Office-Caltech-10 (Gong et al., 2012). The T-SNE is conducted on the test sample features of four clients.

