# OpenReview forum: "A Simple Data Augmentation for Feature Distribution Skewed Federated Learning"
_ICLR.cc/2024/Conference — Submitted to ICLR 2024_

### Official Review · Reviewer_9DEf · 2023-10-28

**Soundness:** 3 good
**Presentation:** 3 good
**Contribution:** 2 fair
**Rating:** 5
**Confidence:** 4

**Summary:**

This paper proposes a new data augmentation method to improve the performance of FL under feature shift. This method can be combined with other existing augmentation methods. The experiments demonstrate that the proposed method achieves better performance.

**Strengths:**

- Addressing the skewed feature distribution problem from a data perspective is interesting and important.
- The proposed method is simple yet seems to be effective in addressing the feature shift problem, while keeping rather high privacy and security of the local data.
- The paper is generally well-written and easy to follow.
- The proposed method can be easily combined with existing methods and elevate their performances.

**Weaknesses:**

- Several existing methods are not compared in the paper: e.g., FedBN [1] and FedWon [2], which also focuses on addressing the feature shift problem. It seems that FedBN achieves better performance than most of the baselines + FedRDN at least in certain domains.
- Only AlexNet is used for evaluating classification tasks.
- The scope of the paper in terms of FL scenario is not clearly explained. Does the method work under cross-silo FL, cross-device FL, or both?

[1] Fedbn: Federated learning on non-iid features via local batch normalization

[2] Is Normalization Indispensable for Multi-domain Federated Learning?

**Questions:**

- How would the proposed method perform using other models, such as ResNet?
- It seems that the reason why the FedRDN can improve performance is not well illustrated in the manuscript.
- Figure 3 in the supplementary is not very intuitive to demonstrate the superiority of FedRDN. Can the author offer more explanation?

---

> ### Author Response · Authors · 2023-11-19
> **Response to reviewer 9DEf**
>
> Thanks for your constructive comments!
>
> > Q1. Several existing methods are not compared in the paper: e.g., FedBN [1] and FedWon [2], which also focuses on addressing the feature shift problem. It seems that FedBN achieves better performance than most of the baselines + FedRDN at least in certain domains.
>
>
> Due to the absence of code release for FedWon,  we compare FedBN with our method on Office-Caltech-10 and DomainNet with the same experimental settings.  In addition, we have included MOON [3] and FPL [4] in the baseline. The results are reported below. we can see that FPL and FedBN can outperform several baselines + FedRDN. However, FedRDN enables some outdated methods to achieve closed or better performance than these SOTA methods.
>
>
> | Dataset | FedBN | MOON | FPL |
> | :-----|:----: |:----: |:----: |
> | Office-Caltech-10 | 70.65 |  63.20 | 71.27 |
> | DomainNet | 43.56 | 42.75  | 44.06 |
>
>
> > Q2. How would the proposed method perform using other models, such as ResNet?
>
>
> Thank you for your valuable suggestion. In response, we have carried out supplementary experiments using ResNet-18 on Office-Caltech-10. The outcomes of the supplementary experiments are provided in the following sections.
> Due to the small size of the dataset, the performance of ResNet18 is not as good as that of AlexNet. However, the results underscore the consistent enhancements attained through our proposed augmentation approach, underscoring its efficacy, adaptability, and resilience.
>
> | Dataset | Network | FedAvg | FedAvg + norm | FedAvg + FedMix | FedAvg + FedRDN |
> | :-----|:----: |:----: |:----: |:----: |:----: |
> | Office-Caltech-10 | AlexNet | 62.51 | 61.46 | 63.59 | 69.80 |
> | Office-Caltech-10 | ResNet-18 | 43.74 | 43.60 | 53.81 | 56.70 |
>
> > Q3. It seems that the reason why the FedRDN can improve performance is not well illustrated in the manuscript.
>
> The effectiveness of our method is owing to learning shared distribution information for all clients, thereby indirectly mitigating the bias of model optimization. Experimental results compared with traditional normalization in Tables 1 and 2,  and cross-site evaluation in Table 3 demonstrated that the effectiveness of our method is not from the pixel-wise re-scaling but from its capacity for biased distribution.
>
> > Q4. Figure 3 in the supplementary is not very intuitive to demonstrate the superiority of FedRDN. Can the author offer more explanation?
>
> In Fig. 3, we employed t-SNE to visualize the feature distribution of the global model for the images belonging to the same category across four different clients. Due to the underlying distributions among these clients, their feature distributions exhibit a shift, referred to as the feature shift phenomenon. Intuitively, as shown in Fig.3 (a),  the features of client 2 (DSLR) are clustered in the bottom-left corner, showing a noticeable shift from the features of other clients. After applying our method, the features of all four clients are uniformly distributed, indicating that the features from these clients are now in a shared feature space. The above result shows that FedRDN can effectively mitigate the feature shift. We will add more explanation in the final version.
>
> > Q5. The scope of the paper in terms of FL scenario is not clearly explained. Does the method work under cross-silo FL, cross-device FL, or both?
>
> Feature distribution skewed FL is a type of heterogeneous FL scenario that focuses on addressing diverse local distributions. Therefore, it can work under both cross-silo and cross-device scenarios, as long as there is feature distribution skew among different clients' data. However, in cross-device scenarios, such as between mobile phones and cloud servers, there may be imbalances in the computational resources of the devices. This issue is beyond the scope of this study.
>
> **References**
>
> [1] FedBN: Federated Learning on Non-IID Features via Local Batch Normalization. ICLR, 2021.
>
> [2] Is Normalization Indispensable for Multi-domain Federated Learning? arxiv, 2023.
>
> [3] Model-Contrastive Federated Learning. CVPR, 2021.
>
> [4] Rethinking Federated Learning With Domain Shift: A Prototype View. CVPR, 2023.

---

> > ### Comment · Reviewer_9DEf · 2023-11-23
> >
> > Thanks for the responses! However, the reviewer still has some concerns.
> >
> > Q1: Do the results on the table represent the original method or the method + FedRDN? If it is the method + FedRDN, what is the performance of the original method? It seems that FedWon[2] also has experiment results on these datasets. Can they be compared directly? Some discussion should be given in the related work session.
> >
> > Q2: The results on ResNet-18 seem to be abnormal. It should not have such a large performance gap with AlexNet. Some hyperparameter tuning may be needed.
> >
> > Q5: Could the authors help demonstrate that the method could work on cross-device FL, under the assumption the computation resources of the device are similar? For example, the total number of clients is 100, and only a small subset of clients are selected.

---

> > > ### Author Response · Authors · 2023-11-23
> > > **Response to Reviewer 9DEf**
> > >
> > > Thanks for your further comments.
> > >
> > > > Q1: Do the results on the table represent the original method or the method + FedRDN? If it is the method + FedRDN, what is the performance of the original method? It seems that FedWon[2] also has experiment results on these datasets. Can they be compared directly? Some discussion should be given in the related work session.
> > >
> > > Thanks for your comments. The results in the table depict the performance of the original method, i.e., without FedRDN. We present the performance of FedBN+FedRDN as follows. Due to client-specific batch normalization (BN) layers, the improvements of FedBN by FedRDN may not be as pronounced as in other methods, but it still shows a certain level of improvements and outperforms other data augmentation methods. Due to differences in experimental setup between our method and FedWon, e.g., epochs, rounds, and optimizer,  we can not directly compare the results shown in the paper of FedWon.  The discussion about FedWon will be included in final version.
> > >
> > >
> > > | Dataset | FedBN | FedBN + norm | FedBN + FedMix | FedBN + FedRDN |
> > > | :-----|:----: |:----: |:----: |:----: |
> > > | Office-Caltech-10 | 70.65 | 69.47 | 70.833 | 71.14 |
> > > | DomainNet | 43.56 | 43.88 | 42.53 | 44.54 |
> > >
> > > > Q2: The results on ResNet-18 seem to be abnormal. It should not have such a large performance gap with AlexNet. Some hyperparameter tuning may be needed.
> > >
> > > For the expriments on ResNet-18, we only change the network and do not alter any other experimental setup. As demonstrated in FedWon, the performance of ResNet-18 is consistently lower to that of AlexNet. The large performance gap with AlexNet may be attributed to the fact that we adopt more epochs and fewer rounds. A larger number of local epochs tends to exacerbate model divergence, leading to a decline in model performance. For larger models, this divergence becomes even more pronounced. Besides, we do not use the learning rate adjustment strategy. This may be also one of the reasons for the performance gap. We will tune the experimental parameters to validate this in final version. More importantly, we employed the same experimental setup for all methods with ResNet-18 to ensure the reliability of the results. Thus, the results are convincing. We will release our code soon for further clarity.
> > >
> > > > Q5. Could the authors help demonstrate that the method could work on cross-device FL, under the assumption the computation resources of the device are similar? For example, the total number of clients is 100, and only a small subset of clients are selected.
> > >
> > > Thanks for your suggestion. Due to the small size of the dataset, with a maximum of only six clients, we are currently unable to validate such experiments. We plan to conduct this verification in future work.

---

### Official Review · Reviewer_rKgT · 2023-10-28

**Soundness:** 3 good
**Presentation:** 3 good
**Contribution:** 2 fair
**Rating:** 5
**Confidence:** 4

**Summary:**

In this paper, a data augmentation approach is proposed to tackle the issue of feature distribution skew in FL. This technique involves the computation and sharing of the mean and standard deviation of data across local client devices. Throughout the training process, data samples are normalized using randomly selected mean and standard deviation values from these stored statistics. This mechanism injects global information, resulting in improved accuracy.

**Strengths:**

**Clarity & Quality**: This paper presents its method in a straightforward manner, substantiated by a series of experiments. The experimental outcomes are presented through tables and figures, facilitating a clear assessment of the efficacy of their approach.

**Originality & Significance**: This paper presents an innovative approach involving the direct application of data statistics to input data. This method proves to be effective in addressing feature distribution skew within FL, resulting in improved accuracy

**Weaknesses:**

1. The paper lacks theoretical analysis and comprehensive explanation. The proposed method needs more elaboration and theoretical support. (Q1, Q2)

2. This study exhibits certain resemblances to FedFA. The approach involving the calculation of mean and standard deviation bears similarity to FedFA, with the key distinction being that this work concentrates on normalizing the input data. In this context, the paper could lack novelty or benefit from more comprehensive comparisons with FedFA.

4. This approach has the potential to cause privacy risks. For instance, if the training data consists of patient information vectors from a specific hospital, sharing the mean and std could compromise the confidentiality of this sensitive patient data.

3. Some details of the experiments are missing. (Q3)

**Questions:**

1. In section 3.2, the author claims that sharing aggregated statistics is like injecting global information. Is there any guarantee or analysis to support that normalizing data using randomly selected mean and standard deviation values is considered equivalent to injecting global information?

2. This method chooses different schemes for training and testing since it says "output results may differ due to the varied statistics chosen". Is this difference significant with varied choices of statistics? Could it work if applying no normalization during the test time?

3. What are the details of the involved datasets? For example, how many clients do they have, how many samples on each client, etc.?

---

> ### Author Response · Authors · 2023-11-19
> **Response to reviewer rKgT**
>
> Thanks a lot for your valuable comments!
>
> > Q1. In section 3.2, the author claims that sharing aggregated statistics is like injecting global information. Is there any guarantee or analysis to support that normalizing data using randomly selected mean and standard deviation values is considered equivalent to injecting global information?
>
> During the training process of FedRDN, we will randomly select a statistic to conduct augmentation for each image. Consequently, after multiple rounds, the number of selections surpasses the quantity of statistics by a substantial margin. This implies that each client leverages the local distribution information from all clients to augment each image, which potentially injects all local information from every client into the client-side training. The results of the cross-site evaluation (Table 3) indicate that our approach significantly improves the generalization of local models. Besides, Fig.3 also demonstrates that our approach has alleviated the feature shift among clients, learning more generalized feature representations.
>
> > Q2. This method chooses different schemes for training and testing since it says "output results may differ due to the varied statistics chosen". Is this difference significant with varied choices of statistics? Could it work if applying no normalization during the test time?
>
> Sorry for confusion. 1) During the testing phase, if we continue to randomly select a statistic, multiple inferences on the same image may yield different results, leading to uncertainty in the test result. Besides, there is no communication between the client and the server. Therefore, it is more practical for the client to choose its own statistic. 2) If normalization is omitted during testing, there will be inconsistency in the data distribution between the training and testing phases, leading to a significant degradation in performance.
>
>
> > Q3. What are the details of the involved datasets? For example, how many clients do they have, how many samples on each client, etc.?
>
> Thanks for your comment. As stated in 'Datasets' section (Sec 4.1), we employ the subsets of each dataset as clients, and we present the detailed results of each client in Table 1 and 2.  The data sizes of the three datasets are shown as below. We will include it in the final version.
>
> | **Office-Caltech-10** | Amazon | Caltech | DSLR | Webcam |
> | :-----|:----: |:----: |:----: |:----: |
> | train | 459 | 538 | 75 | 141 |
> | test | 192 | 225 | 32 | 59 |
>
> | **DomainNet**  | Clipart | Infograph | Painting | Quickdraw | Real | Sketch |
> | :-----|:----: |:----: |:----: |:----: |:----: |:----: |
> | train | 672 | 840 | 791 | 1280 | 1556  | 708 |
> | test | 526  | 657 | 619 | 1000 | 1217 | 554 |
>
> | **ProstateMRI**  | Clipart | Infograph | Painting | Quickdraw | Real | Sketch |
> | :-----|:----: |:----: |:----: |:----: |:----: |:----: |
> | train | 156 | 94 | 280 | 230 | 246  | 105 |
> | test | 52  | 31 | 93 | 76 | 82 | 35 |
>
> > Q4. Difference between our method and FedFA.
>
> Thanks for your comment.  We clarify it in several aspects as follows: 1) Although the process of calculating statistics shares similarities with FedFA's computation process, they lie in their respective focus on input data and features, respectively. Therefore, our approach can be further combined with FedFA to improve its performance. 2) FedFA requires modifying the network structure, which may be constrained in real-world applications. In contrast, our approach requires no modifications to the network, which can be seamlessly integrated into the data augmentation pipeline. Therefore, our method exhibits better generalization and versatility.
>
> > Q5. This approach has the potential to cause privacy risks. For instance, if the training data consists of patient information vectors from a specific hospital, sharing the mean and std could compromise the confidentiality of this sensitive patient data.
>
> In Sec.5, we stated that our method is only applicable to visual tasks. For image data, the statistics ($\mathbb{R}^3$ for RGB images) are privacy-irrelevant information, as they only capture the distribution information. Additionally, they represent the statistics of local datasets and do not contain individual-level information. Therefore, our method is privacy-secure.

---

### Official Review · Reviewer_aJPJ · 2023-10-30

**Soundness:** 3 good
**Presentation:** 2 fair
**Contribution:** 3 good
**Rating:** 6
**Confidence:** 3

**Summary:**

This work aims at tackling feature distribution skewed in FL. To this end, the authors propose a simple yet effective method where the statistics of data are shared across clients to augment local data. Solid experiments are conducted to verify the effectiveness of the proposed method.

**Strengths:**

1. The proposed method is simple yet effective, with relatively high privacy security.

2. Many scenarios are considered for evaluating the proposed method, providing solid experimental evaluation. The experimental results, like performance gain, are promising.

**Weaknesses:**

1. The authors may overlook some related works. The authors claim “few studies pay attention to the data itself”, but many works pay attention to the data itself in FL, such as [1] [2] and [3].

2. It is hard to figure out why “injects the statistics of the dataset from the entire federation into the client’s data” can cause “effectively improve the generalization of features, and thereby mitigate the feature shift problem.” This is the key contribution of this work, but the authors claim it without support. This significantly weakens the contribution of this work.

[1] Federated learning with non-iid data. Zhao et al. 2018
[2] Virtual Homogeneity Learning: Defending against Data Heterogeneity in Federated Learning. Tang et al. 2022
[3] Federated learning via synthetic data. Goetz and Tewari. 2020

**Questions:**

I have several suggestions that may make the work more attractive:

1. I suggest the authors do careful proofreading so that the paper can be more rigorous. For instance, the authors claim that “its (FL model) performance inevitably degrades, while suffering from data heterogeneity”. However, if clients hold iid data, its performance is comparable to the scenario of centralized training.
2. According to the authors’ explanation, it is hard to figure the difference in data heterogeneity and feature shit or feature distribution skewed. I suggest the authors do careful proofreading so that the paper is more readable.
3. All experiments are conducted under the P(X) shifting scenarios. I suggest the authors report more results on the scenario of P(Y) shifts, which may make the work more attractive (do not have much stress, as it is just a suggestion).
4. Detailed descriptions for Figure 3 will make the motivation and conclusion more clear.

---

> ### Author Response · Authors · 2023-11-19
> **Response to reviewer aJPJ**
>
> Thanks a lot for your constructive comments!
>
> > Q1. The authors may overlook some related works. The authors claim "few studies pay attention to the data itself", but many works pay attention to the data itself in FL, such as [1] [2] and [3].
>
> Literature [1], [2], and [3] focus on label distribution skew, which are different from the issue in this paper.
>
> > Q2. It is hard to figure out why "injects the statistics of the dataset from the entire federation into the client’s data"  can cause "effectively improve the generalization of features, and thereby mitigate the feature shift problem."  This is the key contribution of this work, but the authors claim it without support. This significantly weakens the contribution of this work.
>
> During the training process of FedRDN, we will randomly select a statistic to conduct augmentation for each image. Consequently, after multiple rounds, the number of selections surpasses the quantity of statistics by a substantial margin. This implies that each client leverages the local distribution information from all clients to augment each image, which potentially injects all local information from every client into the client-side training. The results of the cross-site evaluation (Table 3) indicate that our approach significantly improves the generalization of local models. Besides, Fig.3 also demonstrates that our approach has alleviated the feature shift among clients, learning more generalized feature representations.
>
> > Q3. I suggest the authors do careful proofreading so that the paper can be more rigorous. For instance, the authors claim that "its (FL model) performance inevitably degrades, while suffering from data heterogeneity." However, if clients hold iid data, its performance is comparable to the scenario of centralized training.
>
> Sorry for confusion. We will carefully proofread our paper.
>
> > Q4. According to the authors' explanation, it is hard to figure the difference in data heterogeneity and feature shit or feature distribution skewed. I suggest the authors do careful proofreading so that the paper is more readable.
>
>
> Sorry for confusion. We will maintain consistency in words and carefully revise it in the final version.
>
>
> > Q5. All experiments are conducted under the P(X) shifting scenarios. I suggest the authors report more results on the scenario of P(Y) shifts, which may make the work more attractive (do not have much stress, as it is just a suggestion).
>
> P(Y) shifting is label distribution skew, which is different from the issue in this paper. This work focuses on the feature distribution skew, i.e., P(X) shifting. The details of this problem can be seen in Sec 3.1.
>
> > Q6. Detailed descriptions for Figure 3 will make the motivation and conclusion more clear.
>
> In Fig. 3, we employed t-SNE to visualize the feature distribution of images belonging to the same category across four different clients. Due to the different distributions among these clients, their feature distributions exhibit a shift, referred to as the feature shift phenomenon. Intuitively, as shown in Fig.3 (a),  the features of client 2 (DSLR) are clustered in the bottom-left corner, showing a noticeable shift from the features of other clients. After applying our method, the features of all four clients are uniformly distributed, indicating that the features from these clients are now in a shared feature space. The above result shows that FedRDN can effectively mitigate the feature shift. We will add more explanation in the final version.
>
> **References**
>
> [1] Federated learning with non-iid data. Zhao et al. 2018
>
> [2] Virtual Homogeneity Learning: Defending against Data Heterogeneity in Federated Learning. Tang et al. 2022
>
> [3] Federated learning via synthetic data. Goetz and Tewari. 2020

---

> > ### Comment · Reviewer_aJPJ · 2023-11-23
> > **Re: response**
> >
> > Thanks for the detailed response. After reading the response and other reviewers' comments, I think the paper is borderline. I am going to keep my original score.

---

### Official Review · Reviewer_BGPc · 2023-11-03

**Soundness:** 3 good
**Presentation:** 3 good
**Contribution:** 3 good
**Rating:** 5
**Confidence:** 3

**Summary:**

This paper discusses the problem of feature distribution skew in federated learning (FL) and proposes a data augmentation technique called FedRDN to mitigate this issue. The main challenge in FL is data heterogeneity, which leads to feature shift due to different underlying distributions of local datasets. While previous studies have focused on addressing this issue through model optimization or aggregation, few have paid attention to the data itself. FedRDN addresses this by randomly injecting the statistics of the dataset from the entire federation into the client's data, improving the generalization of features and mitigating feature shift. The method is simple, effective, and can be seamlessly integrated into the data augmentation flow. Experimental results demonstrate its scalability and generalizability. The document also provides a summary of related work in FL with statistical heterogeneity and data augmentation techniques.

**Strengths:**

1. This approach is different from previous methods and focuses on mitigating the feature shift at the input-data level.

2. The paper is generally well-written and clear in presenting the problem, proposed approach, and experimental results.

3. Experiments are conducted on multiple datasets.

**Weaknesses:**

1. The literature review appears to be incomplete or lacks recent research contributions.

2. Improved paragraph transitions and organization are required.

3. The presentation needs improvement.

**Questions:**

1. I am not that familiar with the skewed FL scenario, could you explain more about it?
2. In Section 1, the effectiveness of data augmentation for FL at the input level has been mentioned times, how to improve it in this paper?
3. In term of data augmentation, what are the differences between previous methods and the proposed method?
4. In Eq.(6), the obtained image are equipped with global information, but how to measure the contribution of multiple distributions?
5. The paper will be more attractive if state-of-the-art methods are included in experiments for comparison.
6. For the purpose of reproducibility, it would be better to provide the code and datasets.

---

> ### Author Response · Authors · 2023-11-19
> **Response to reviewer BGPc**
>
> Thanks a lot for your constructive comments!
>
> > w1. Literature review is incomplete.
>
> We carefully reviewed recent literatures, which are related to the feature distribution skewed FL. The following content will be included to the final version.
>
> *'More recently, FedPCL [5] employed a pre-trained model to reduce the number of learnable parameters and applied prototype-wise contrastive learning to regularize feature representation learning across different clients. This offers an effective solution for training large models in the feature distribution skewed federated learning (FL) scenario. Motivated by prototype learning, FPL [6] utilized clustering to acquire unbiased class prototypes and then alleviated feature shift through prototype and local embedding alignment. In contrast to the aforementioned methods, ADCOL [7] introduced a novel adversarial collaborative learning approach to mitigate feature shift, replacing the model-averaging scheme with adversarial learning.'*
>
> > w2 & w3. Problem of writing.
>
> We will revise the language of manuscript and invite native speakers for proofreading.
>
> > Q1. I am not that familiar with the skewed FL scenario, could you explain more about it?
>
> Feature distribution skew [1] is a fundamental challenge in federated learning. As federated learning typically involves multiple discrete clients, data collected by different clients unavoidably leads to distinct underlying distributions [3]. For client $k$, the underlying data distribution $P_k(x,y)$ can be rewritten as $P_k(y|x)P_k(x)$, and $P_k(x)$ varies across clients while  $P_k(y|x)$ is consistent for all clients. For instance, different hospitals possess MRI images scanned by different devices, and different phones store images of different styles (such as cartoon images and natural images). In summary, feature distribution skewed federated learning (FL) typically focuses on multi-domain data. Due to variations in the distributions of different data domains, it results in biased feature distributions among different local models [2, 7].
>
>
> > Q2&Q3. Explanation of data augmentation.
>
> Sorry for confusion. There may be some misunderstandings. In previous FL research, there is no input-level data augmentation method. To the best of our knowledge, **this paper is the first work to explore the input-level data augmentation for feature distribution skewed FL**. Traditional input-level data augmentation methods in centralized learning often struggle to effectively address the feature shift issue stemming from distributional differences among local datasets. In this work, we focus on extending the traditional normalization operation to handle the feature distribution skewed FL scenario. To achieve this, we transfer distribution characteristics of all clients, i.e., mean and std, and utilize them to augment the images of each client. Cross-site evaluation (Table 3) and feature distribution visualization (Fig.3)) have demonstrated our method can effectively mitigate feature shift, thereby significantly improve the peformance (Tables 1 and 2).
>
> > Q4. In Eq.(6), the obtained image are equipped with global information, but how to measure the contribution of multiple distributions?
>
> Sorry for confusion. In Eq.(6), we did not aggregate the statistics from all clients to get gloabl satistic but randomly selected one from them for augmentation. Therefore, there is no need to measure the contribution of each client here.
>
> > Q5. The paper will be more attractive if state-of-the-art methods are included in experiments for comparison.
>
> Thanks for your comment. Based on your suggestion, we compared more baselines (FedBN [2], MOON [4] and FPL [6]) on Office-Caltech-10 and DomainNet with the same experimental settings. The results are repoted as below. we can see that FPL and FedBN can outperforms several baselines + FedRDN. However, FedRDN enables some outdated methods to achieve closed or better performance than these SOTA methods.
>
> | Dataset | FedBN | MOON | FPL |
> | :-----|:----: |:----: |:----: |
> | Office-Caltech-10 | 70.65 |  63.20 | 71.27 |
> | DomainNet | 43.56 | 42.75  | 44.06 |
>
>
> > Q6. For the purpose of reproducibility, it would be better to provide the code and datasets.
>
> Thanks for your suggestion. The dataset is publicly available and the code will be released.
>
> **References**
>
> [1] Federated Learning on Non-IID Data Silos: An Experimental Study. ICDE, 2022.
>
> [2] FedBN: Federated Learning on Non-IID Features via Local Batch Normalization. ICLR, 2021.
>
> [3] FedFA: Federated Feature Augmentation. ICLR, 2023.
>
> [4] Model-Contrastive Federated Learning. CVPR, 2021.
>
> [5] Federated Learning from Pre-Trained Models: A Contrastive Learning Approach. NeurIPS, 2022.
>
> [6] Rethinking Federated Learning With Domain Shift: A Prototype View. CVPR, 2023.
>
> [7] Adversarial Collaborative Learning on Non-IID Features. ICML, 2023.

---

### Meta-Review · Area_Chair_e6Wr · 2023-12-20

**Metareview:**

Although all reviewers found this paper interesting, they do point out several opportunities for improving, centered around novelty, theoretical understanding and analysis, and potential empirical results to further support some of the statements.

**Justification For Why Not Higher Score:**

There are several opportunities for improving, centered around novelty, theoretical understanding and analysis, and potential empirical results to further support some of the statements.

**Justification For Why Not Lower Score:**

N/A

---

### Decision · Program_Chairs · 2024-01-16

Reject